# A Numerical Study on the Flap Side-Edge Noise Reduction Using Passive Blowing Air Concept

**Yingzhe Zhang** [1,2,3], **Baohong Bai** [2,3,*], **Dakai Lin** [2,3] **and Peiqing Liu** [1]

1 School of Aeronautic Science and Engineering, Beihang University, Beijing 100191, China
2 Beijing Aircraft Technology Research Institute of Commercial Aircraft Corporation of China, Ltd., Beijing 102211, China
3 Beijing Key Laboratory of Simulation Technology for Civil Aircraft Design, Beijing 102211, China
\* Correspondence: bobaohong@comac.cc

**Abstract:** The flap side-edge is a vital contributor to airframe noise. In this study, we propose a novel flap side-edge noise reduction method based on the concept of active blowing air. A long slot is opened from the flap's lower surface to the tip surface to induce a secondary jet flow, which is driven by the local pressure difference between the flap's lower surface and the tip surface. The unsteady flow field around the flap side-edge was computed by the lattice Boltzmann solver PowerFLOW, and the far-field noise was predicted by the FW-H equation. It is demonstrated that the dominant features of the flap side-edge flow are the double vortex structures, and the new passive blowing air reduction method can achieve about 3.3 dB noise reduction. Moreover, the underlying noise reduction mechanism has been analyzed and revealed. It is shown that the secondary jet flow from the long slot on the flap side-edge would dissipate the flap side-edge vortex and displace the flap side-edge vortical structure away from the flap surface, thus resulting in a decrease in the pressure fluctuations on the flap side-edge surface. As a result, the flap side-edge noise was reduced. In contrast to the current active air blowing technique, the newly proposed blowing air technique is passive and quite simple and does not require an extra air source or control system. This novel flap side-edge noise reduction technology provides a new flow control strategy and noise reduction methodology and can be further optimized.

**Keywords:** aeroacoustics; airframe noise; flap side-edge noise





## 1. Introduction

In the past few decades, significant noise reduction of the propulsion systems of commercial aircraft has been achieved through aeroacoustic research, so that the aero-engines are no longer the only important aircraft noise source. Especially at the aircraft landing stage, airframe noise has become the dominant aircraft noise source when the aero-engines are operated at reduced thrust, the landing gear are deployed, and the high-lift systems are extended. As a result, more research has been conducted on the understanding and reduction of airframe noise components such as landing gear, slats, and flaps [1,2].

It is well known that the flap side-edge is one of the major airframe noise sources [2–5]. The flaps are designed to generate aircraft lift at the takeoff and approach stages. Unfortunately, this design also led to unstable flow features around the flap side-edge regions that produce intense noise [6]. The dominant feature of the flow in the flap side-edge regions is the formation and evolution of double-vortex structures, which have been observed and investigated both in numerical simulation and experiment measurement [7–12]. The formation of these double-vortex structures results from the inevitable flow separation of the spanwise cross flow, which is due to the pressure differences between the pressure and suction surfaces of the flap. The boundary layer on the flap pressure surface separates at the sharp leading edge of the flap and rolls up to form a strong vortex.

The adjacent outer flow reattaches to the flap tip surface and forms a thin boundary layer. The newly formed boundary layer separates at the sharp top edge and forms a weak vortex on the flap suction surface. Both vortices are convecting downstream and growing continuously in strength and size. Eventually, both vortices merge into a single dominant streamwise vortex on the aft part of the flap suction surface [13,14]. It indicates that the flap side-edge flow is highly unstable and inherently unsteady, providing potential noise sources from flow fluctuations that may escape to the far field as sound.

In recent years, two dominant noise generation mechanisms have been identified for the flap side-edge noise [14–16]. The first mechanism for generating noise is the interaction between the shear layer fluctuations and the flap side-edge surface. The second mechanism for generating noise is the amplification of the unstable modes of the vortex structure by the shear layer fluctuations. Both shear layer fluctuations and vortex instabilities lead to high-intensity pressure fluctuations on the solid surface of the flap side-edge and thus to sound radiation into the far field. On this basis, Guo [16] developed a physics-based two-source flap side-edge noise prediction model, in which two noise sources correspond to two different noise generation mechanisms. Recently, Bai et al. [17] identified two sources of flap side-edge noise based on the phased microphone array. It is demonstrated that two different regions at the flap side-edge are responsible for the low and high frequencies of the noise source. The aft half-chord of the flap side-edge mainly produces the low-frequency noise. In contrast, the forward half-chord of the flap side-edge generates the high-frequency noise.

Based on the understanding of the flap side-edge flow features and noise generation mechanisms, many flow control techniques and noise reduction methods have been proposed and investigated by experiments and computations. In general, these noise reduction methods can be classified into passive noise reduction methods and active noise reduction methods. The passive reduction methods of the flap side-edge noise include the changing of the flap side-edge shape and materials and the elimination of the gap between the flap side-edge and main wing.

The changes in flap side-edge noise caused by rounding the flap side-edges have been numerically and experimentally investigated [18]. Compared with the baseline flap configuration, which has a rectangular cross section, the "LowerRound" flap side-edge, which has a rounded edge at the bottom edge and a sharp edge at the top edge, demonstrated the most remarkable noise reduction, while the "UpperRound" flap side-edge, which has the opposite geometry, increased the flap side-edge noise level. Furthermore, they proposed a protruding-type flap side-edge using a similar noise reduction mechanism with a "LowerRound" flap side-edge shape to weaken the flap side-edge vortex rolled up from the bottom edge of the flap side-edge, which is referred to as a "Protruding Device". Numerical simulation results demonstrated that the "Protruding Device" flap side-edge shape shows a greater noise reduction effect than that of the "LowerRound" at the middle frequency range.

Flap side-edge fences are typically flat plates attached to the pressure surfaces of the flaps such that the plates protrude into the flow on the pressure side of the flaps. These fences would modify the local flow structures to reduce flap side-edge noise [19]. It is demonstrated in both simplified flaps and realistic aircraft configurations [20,21] that these fences are effective in reducing flap side-edge noise. Typically, flap side-edge fences can achieve a noise reduction of up to 4 dB in the middle to high frequency domain.

Another effective approach to suppress flap side-edge noise is to eliminate the flap side-edge by using continuous mold-line link (CML) technology [22–24]. The goal of the CML technology is to prevent the formation of strong and concentrated vortical structures and break up the single strong vortex into a spanwise distribution of weaker vortices. Streett et al. [22] studied the CML concept in a high-lift system configuration. Their experiment results indicated that a large reduction of the flap side-edge noise was achieved when compared to that obtained from a baseline flap side-edge.

Porous flap side-edge treatment [25,26] is another promising passive method for flap side-edge noise reduction. It was founded that the flap side-edge made of porous materials would decrease the noise. This is mainly attributed to the fact that the porous flap side-edge modifies the flap side-edge vortex initiation and roll-up processes. As a result, the flap side-edge vortical structures are significantly weakened. In addition, numerical simulations confirmed that the porous treatment would achieve flap side-edge noise reduction without compromising the aerodynamic effectiveness of the flap.

The active flow control and noise reduction methods include plasma and air blowing/suction. It has been widely demonstrated that the plasma technique is able to modify the local flow by introducing extra momentum [27]. The combination of passive flap side-edge shaping in conjunction with plasma-induced air blowing could weaken and displace flap side-edge vortical structures away from the flap side-edge. As a result, the flap side-edge noise was reduced. Koop et al. [28] experimentally demonstrated flap side-edge noise reduction by air blowing. Air was blown into the flap side-edge vortical structures through a series of small round orifices located along the flap bottom and top corners to displace or destroy the vortical structures and thus reduce the flap side-edge noise. PIV (Particle Image Velocimetry) measurements with air blowing show that the flap side-edge vortical structures can be almost completely dispersed and that the maximum vorticity in the vortex core is significantly reduced. As a result, a noise reduction of 3–4 dB was achieved. However, this active blowing of air is quite complex because it requires an extra air source and control system.

Considering the local pressure difference between the lower surface and the tip surface of the flap, it is possible to employ the local pressure difference to produce a secondary jet flow to dissipate the flap side-edge vortex and force the flap side-edge vortex away from the flap side-edge surface. Therefore, a novel flap side-edge noise reduction method based on the concept of blowing air is proposed in this study. Its noise reduction is evaluated numerically. Section 2 provides a brief introduction to the lattice Boltzmann solver and a description of the simulation setup. The simulation results are compared and discussed in Section 3. Conclusions are made in the last section.

## 2. Computational Methods and Setup

### 2.1. Computational Method

In this study, the commercial solver PowerFLOW was employed to compute the unsteady flow field around the flap side-edge. The commercial solver PowerFLOW is based on the lattice Boltzmann method (LBM). The LBM solver PowerFLOW has been widely used for studying the aeroacoustics of a turbofan model [29–31], landing gear [32], slat [33], and flap side-edge [34].

The LBM is derived from Boltzmann's kinetic theory, which describes a fluid as a collection of particles that continuously evolve towards a thermodynamic equilibrium state. The state of the particles is tracked using a probability distribution function. This governing equation is the Boltzmann transport equation (BTE).

The BTE is solved on a lattice of cubic elements, which are called voxels. Currently, the solver PowerFLOW implements a variable resolution scheme so that the computational domain can be divided into several regions with different voxel sizes. The voxel resolution commonly varies by a factor of two between adjacent regions. When the voxels intersect with solid bodies, planar surfaces are created along the solid-fluid boundaries. The solution computed by the LBM method is inherently unsteady and compressible. Furthermore, the scheme has low dissipation and dispersion properties, which allow for acoustic phenomena to be resolved in the computational domain. This indicates that the LBM solver is able to directly simulate the noise generation and propagation processes. The LBM is typically solved on cartesian grids, so that the computational mesh can be generated for extremely complex geometries. The LBM solver PowerFLOW employs a very large eddy simulation (VLES) approach to turbulence modeling and uses an extended wall function model on a no-slip wall to account for pressure gradients. Nevertheless, the computational cost associated

with extending a high-resolution voxel region into the acoustic far field still remains too expensive. Hence, the far-field acoustic pressure is obtained with the Ffowcs-Williams and Hawkings (FW-H) method [35].

### 2.2. Flap Side-Edge Geometry

In this study, the baseline flap side-edge model is displayed in Figure 1. Figure 1a,b show the baseline flap geometry from the side view and the bottom view, respectively. It can be clearly seen that the baseline flap has a rectangle-shaped side edge with a blunt trailing edge. The chord of the flap is about 0.68 m, and the thickness of the flap is about 0.068 m. The deployment angle of the flap is 30 degrees.

In order to reduce the flap side-edge-generated noise, we propose a reduction technology for flap side-edge noise in this study. Figure 2a,b show the new flap geometry from the side view and the bottom view, respectively. It can be clearly seen that a long slot is produced from the flap side-edge lower surface to the tip surface. The inlet of the slot is on the bottom surface, and the outlet of the slot is on the tip surface. The area of the inlet is about ten times the area of the outlet. This design is intended to accelerate the flow past the slot. The chordwise position of the long slot is from the 5% chord to the 60% chord. The inlet and outlet of the slot have the shape of a rectangle. The outlet of the slot is placed at the center of the tip surface. The new flap geometry with the slot is referred to as "slotted flap side-edge" in this work.

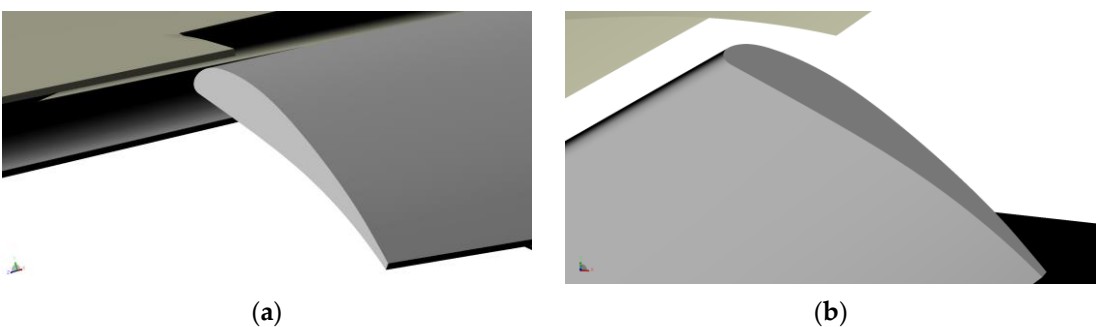

|  (a)  |  (b)  |

**Figure 1.** Geometry around the baseline flap side-edge. (**a**) Side view; (**b**) bottom view.

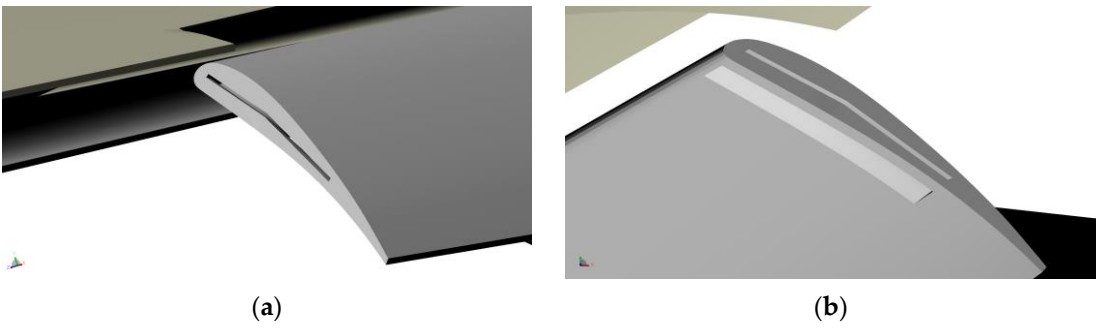

|  (a)  |  (b)  |

**Figure 2.** Geometry around the slotted flap side-edge. (**a**) Side view; (**b**) bottom view.

### 2.3. Computational Domain and Computational Mesh

At the aircraft landing stage, the flap is deployed at its maximum deployment angle, and the flap produces the maximum lift force. As a result, the flap side-edge vortex is much stronger, and thus the flap side-edge radiates the maximum noise. Therefore, the flap configuration at the landing stage was selected for the numerical simulation in this study.

In this study, the flap was investigated in the landing configuration. The angle of attack (AoA) is 5.4 degrees, and the incoming flow Mach number is 0.277. The velocity boundary condition was applied to the inlet boundary, and the far-field pressure boundary condition was employed for the outlet boundary. The computational domain is shown in

Figure 3. The inlet is denoted by the green surface, and the red surface represents the outlet; the other four surfaces were set as the solid slip wall boundary condition.

The reference pressure and reference velocity applied the realistic parameters in flight. The cartesian mesh shown in Figure 4 was generated for the baseline flap side-edge computational model. The finest mesh was distributed around the flap side-edge, and the minimum size of the computation mesh is 0.5 mm. The total number of computation meshes is about 30 million. Figure 4 shows the cross-sectional views of the computational grid for the baseline flap side-edge configuration. An identical setup was made for the computational mesh of the slotted flap side-edge configuration.

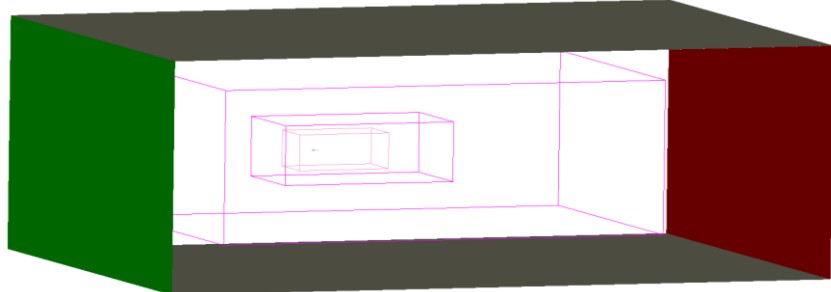

**Figure 3.** Computational domain.

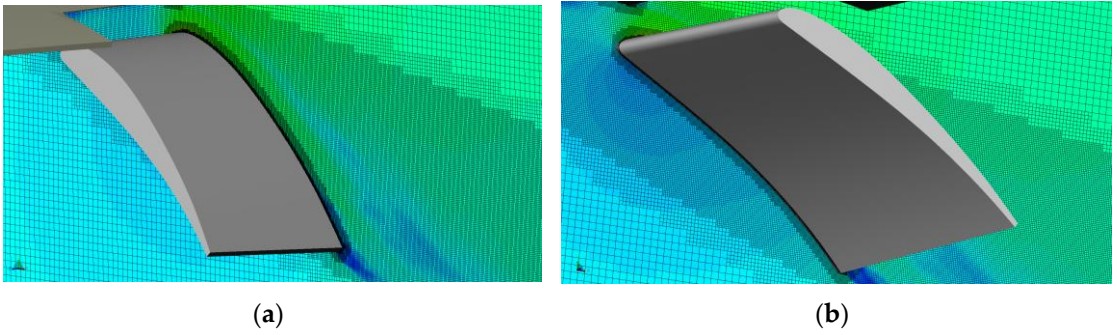

(**a**)　　　　　　　　　　　　　　　　(**b**)

**Figure 4.** Computational mesh around the flap side-edge. (**a**) Side view; (**b**) bottom view.

*2.4. Grid-Convergence Study*

A grid-convergence study of the simulation results is carried out using three different mesh resolutions, namely coarse, medium, and fine, with a refinement ratio of $\sqrt{2}$. The simulation settings are reported in Table 1. The grid resolution level is defined using the number of voxels assigned over the flap chord length ($c$).

**Table 1.** Simulation domain specification for grid convergence study.

| Grid Type | Resolution (Voxels/$c$) | Voxel Count (Millions) |
|:---:|:---:|:---:|
| Coarse | 971 | 10.93 |
| Medium | 1360 | 30.10 |
| Fine | 1923 | 87.46 |

The effect of varying the grid resolution level on the mean flow field is evaluated using the time-averaged aerodynamic force coefficients. Figure 5 shows the variation of the lift coefficients and drag coefficients with the number of voxels in the simulations for the baseline flap side-edge configuration. The lift coefficient and the drag coefficient approach convergence as the number of voxels is increased, such that the differences in the lift coefficient and the drag coefficient between the values for the fine and medium grid resolution levels are 0.49% and 0.36% for the baseline flap side-edge, respectively.

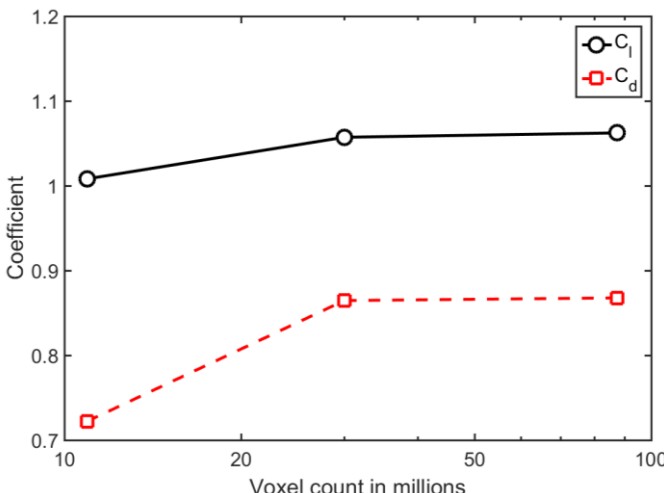

**Figure 5.** The variation of the lift coefficients ($C_l$) and drag coefficients ($C_d$) produced by the baseline flap side-edge against the grid resolution level.

The influence of the grid resolution on the sound emission characteristics of the flap side-edge configuration is evaluated using the sound power level (PWL) spectrum. The source power level was estimated by integrating the far-field noise intensity over a fictitious spherical surface with a radius of 6.8 m (i.e., 10 times the flap chord length) centered at the flap tip surface. A total of 648 measurement points is distributed on the surface with an increment of 10 degrees in both azimuthal and polar directions. The variations of the PWL spectra with the grid resolution level are depicted in Figure 6, where converging trends can be observed for the baseline flap side-edge configuration. The difference in the overall PWL (OPWL) is 0.8 dB between the fine grid and the medium grid.

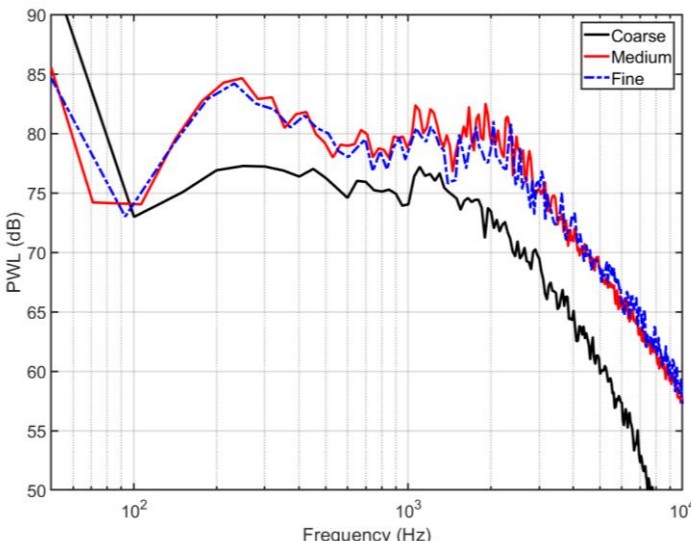

**Figure 6.** The variation of the sound power level (PWL) of the baseline flap side-edge against the grid resolution level.

## 3. Computational Results and Analysis

In this section, the flow field results and far-field noise are given and compared between the baseline flap side-edge and the slotted flap side-edge. The analysis of the flow field is completed.

### 3.1. Analysis of the Flow Field

In this study, the averaged pressure distribution around the flap side-edge is shown in Figure 7. Figure 7a–c compares the averaged pressure distribution between the baseline flap side-edge and the slotted flap side-edge from different views. The averaged pressure distributions for the baseline flap side-edge and the slotted flap side-edge are shown in the top figure and bottom figure, respectively. It can be observed that the maximum difference between the averaged pressure distribution appears around the flap tip surface. Away from the flap tip surface, the averaged pressure distributions are nearly identical. The minimum averaged pressure is located at the half chord for the baseline flap side-edge. This is where the vortex merging happened. Compared with the baseline flap side-edge, the averaged pressure for the slotted flap side-edge is distributed quite evenly, as shown in Figure 7. This indicates that the secondary jet flow from the long slot would dissipate the side-edge vortex.

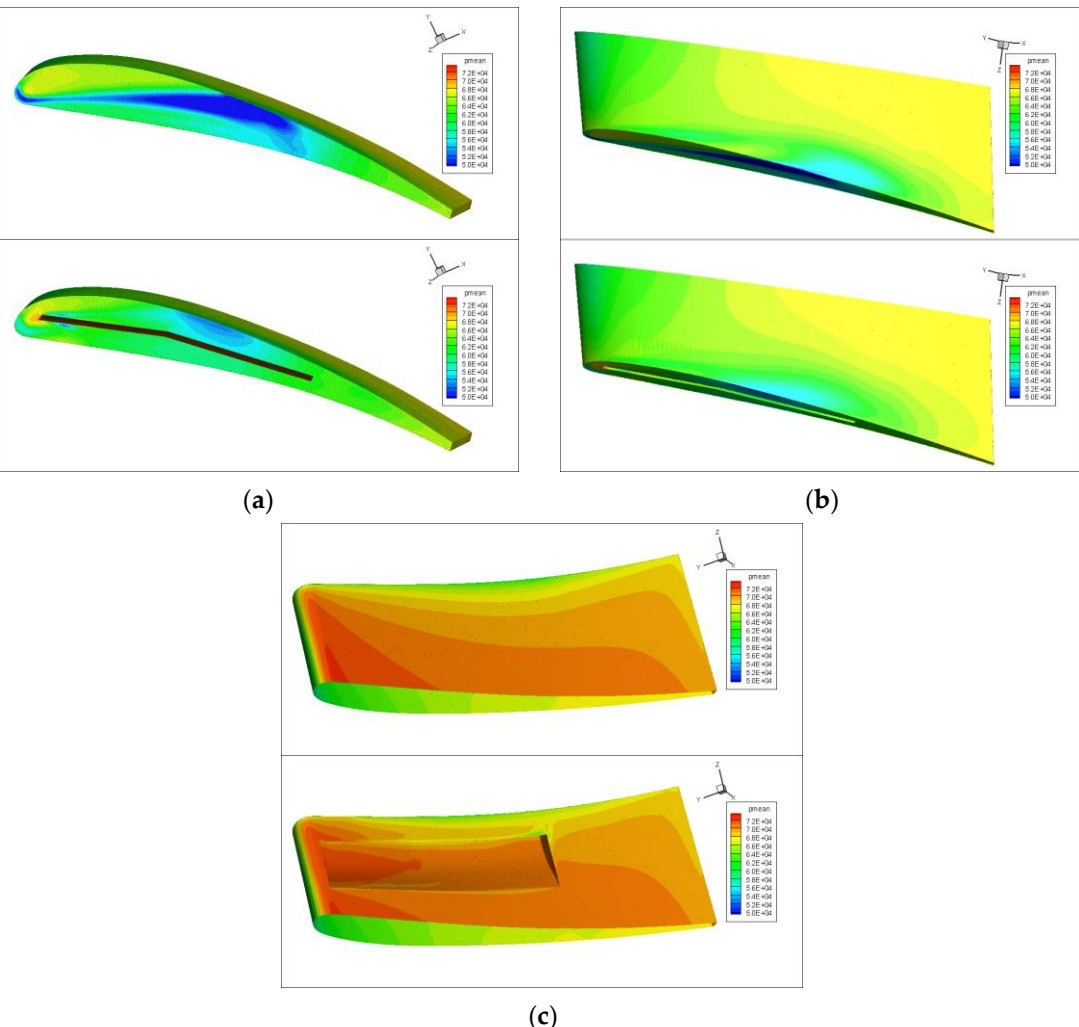

**Figure 7.** The comparison of the averaged pressure distribution around the flap side-edge between the baseline flap side-edge and slotted flap side-edge. (**a**) Side view; (**b**) top view; (**c**) bottom view.

The comparison of the aerodynamic performance is reported in Table 2, which lists the lift coefficients, the drag coefficients, and the lift-to-drag ratio. It can be seen that the slotted flap side-edge has slightly decreased the lift coefficient and drag coefficient. The decrease in lift coefficients is mainly caused by the decrease in the pressure difference between the lower surface and the upper surface. The decrease in drag coefficient can be attributed to the suppression of the flap side-edge vortical structures. In fact, the difference in the

lift-to-drag ratio is 0.7% between the baseline flap side-edge and the slotted flap side-edge. This indicates that the newly proposed flap side-edge noise reduction method has little influence on aerodynamic performance.

**Table 2.** The comparison of aerodynamics performance between the baseline flap side-edge and slotted flap side-edge.

| Aerodynamic Performance | Baseline Flap Side-Edge | Slotted Flap Side-Edge |
| --- | --- | --- |
| Lift coefficient | 1.0571 | 1.0014 |
| Drag coefficient | 0.8646 | 0.8248 |
| Lift-to-drag ratio | 1.2227 | 1.2140 |

The RMS (root mean square) pressure fluctuations were compared between the baseline flap side-edge and the slotted flap side-edge in Figure 8. Figure 8a–c shows the comparison from the side, top, and bottom views, respectively. The RMS pressure fluctuation for the baseline flap side-edge is shown in the top figure, and the RMS pressure fluctuation for the slotted flap side-edge is shown in the bottom figure. For the baseline flap side-edge, the maximum RMS pressure fluctuations are located at the half-chord of the tip surface, where the double vortex merged. In contrast, the slotted flap side-edge decreases the maximum RMS pressure fluctuations. Away from the flap tip surface, the RMS pressure fluctuations are almost identical for the baseline flap side-edge and slotted flap side-edge. It is observed that the RMS pressure fluctuations are significantly reduced by the slotted flap side-edge, which would also decrease the vibration of the flap. As a result, the safety of the flap would increase.

To analyze the effect of the secondary jet flow on the flap side-edge vortical structures, the instantaneous velocity at a cross-section plane around the baseline flap side-edge and the slotted flap side-edge is shown in Figure 9. We can see clearly from Figure 9a that there is a strong vortex around the baseline flap side-edge. This is mainly caused by the pressure difference between the lower surface and the upper surface of the flap. Meanwhile, this roll-up side-edge vortex would induce high pressure fluctuations on the flap side-edge surface, which is a major noise contributor. Figure 9b displays the instantaneous velocity around the slotted flap side-edge. A secondary jet flow is driven by the local pressure difference in the slot. The velocity of the secondary jet flow in the slot increases from the inlet to the outlet. Compared with the baseline flap side-edge as shown in Figure 9a, the velocity around the slotted flap side-edge is quite low, as shown in Figure 9b. The instantaneous velocity around the flap side-edge at a cross-sectional plane clearly demonstrates that the secondary jet flow from the long slot would dissipate the original flap side-edge vortex, leading to a decrease in the pressure fluctuations on the flap surfaces.

Figure 10 displays the vortical structure iso-surface of the Q-criterion, colored by the velocity magnitude, and compares the vortical structures between the baseline flap side-edge and slotted flap side-edge. It can be clearly seen that double vortexes are generated at the flap side-edges. One vortex is generated from the lower side-edge, and another originates from the upper side-edge. Compared with the vortex from the upper side-edge, the vortex from the lower side-edge is much stronger. Both vortexes move downstream independently. Until the half-chord location, the merging of both vortexes had not happened. Compared with the vorticity around the baseline flap side-edge, as illustrated in Figure 10a, the vorticity around the slotted flap side-edge has several different characteristics. The secondary jet flow from the long slot forced the flap side–edge vortex away from the flap surface, thus weakening the interaction between the flap side-edge vortexes and the flap side-edge surface. On the other hand, the secondary jet flow from the long slot would dissipate the flap side-edge vortical structures, which reduces the strength of the vorticity. Both effects would be beneficial for the flap side-edge noise reduction.

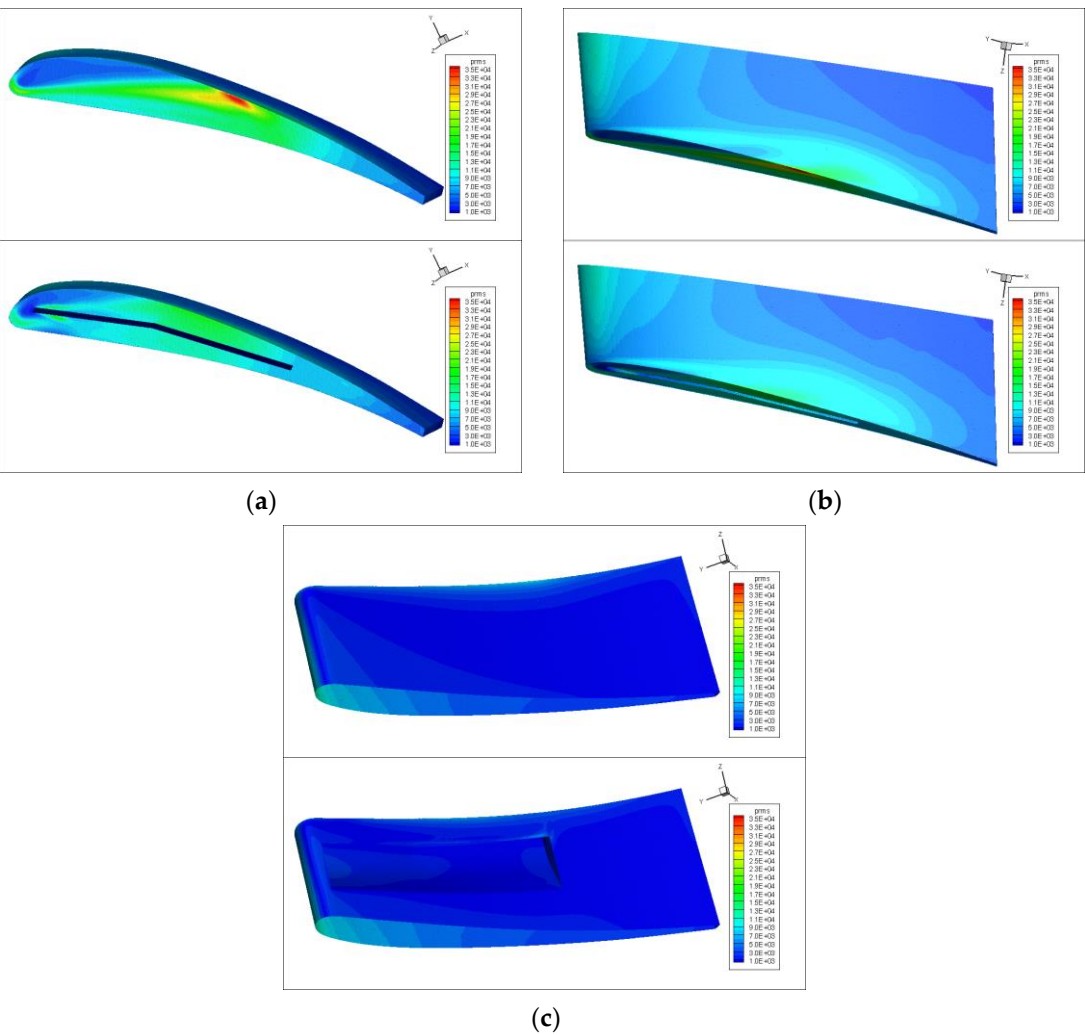

(**a**)                                                          (**b**)

(**c**)

**Figure 8.** The comparison of the RMS pressure distribution around the flap side-edge between the baseline flap side-edge and the slotted flap side-edge. (**a**) Side view; (**b**) top view; (**c**) bottom view.

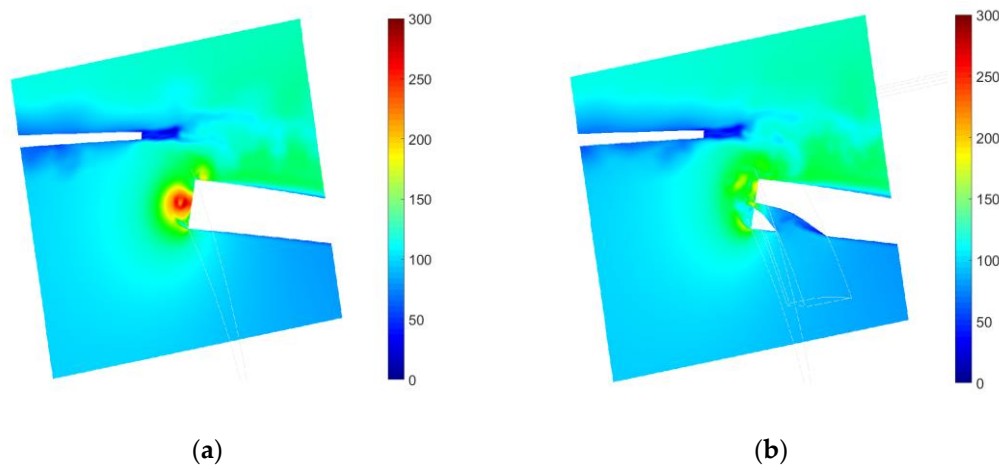

(**a**)                                                          (**b**)

**Figure 9.** The comparison of the instantaneous velocity around the flap side-edge at a cross-section plane. (**a**) Baseline flap side-edge; (**b**) slotted flap side-edge.

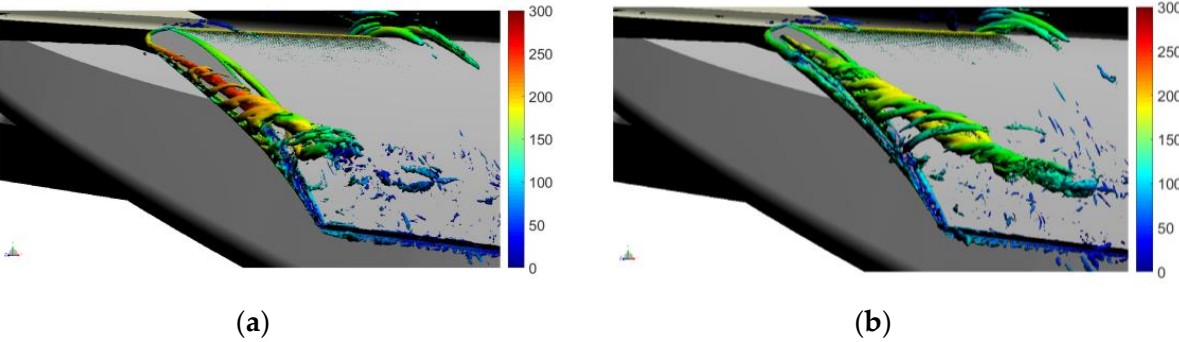

**(a)**        **(b)**

**Figure 10.** Iso-surface of the Q-criterion, colored by the velocity magnitude. (**a**) Baseline flap side-edge; (**b**) slotted flap side-edge.

It is well known that surface pressure fluctuations are the major noise sources. Therefore, the pressure fluctuations are investigated and compared between the baseline flap side-edge and the slotted flap side-edge. Four different virtual sensors are placed at the flap side-edge surface. Figure 11 displays the positions of four different sensors on the flap side-edge surface. P1 is located at the leading edge and tip surface of the flap, where a vortex originates from the lower side-edge of the flap. The P2 sensor is placed at the leading edge and upper surface of the flap, where a vortex originates from the upper side-edge. P3 is placed at the position where the double vortex merging happened. P4 is placed at the trailing edge and the upper surface of the flap. It can be seen that the four different sensors are located at different flow regimes.

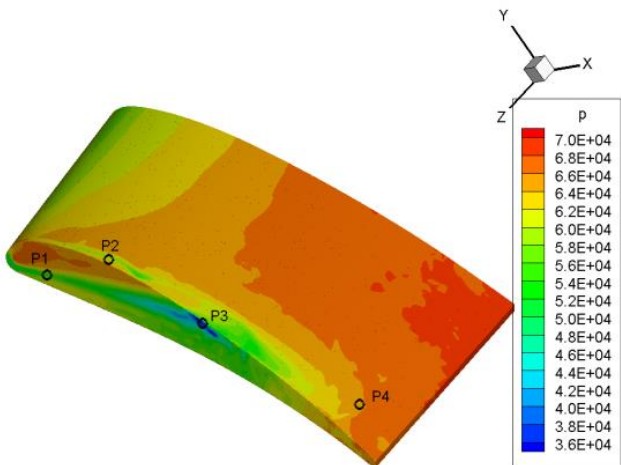

**Figure 11.** The positions of four different sensors on the flap side-edge surface.

The comparisons between the baseline flap side-edge and slotted flap side-edge at the four different sensors are given in Figure 12. It can be seen that the surface pressure spectrum at the P1 sensor for the baseline flap side-edge is higher than that of the slotted flap side-edge in the low-frequency range. This indicates that the long slot changes the lower vortex development. For the P2 sensor, the difference in the pressure spectra at the P2 sensor is small between the baseline flap side-edge and the slotted flap side-edge. This indicates that the secondary jet flow from the long jet has a minor influence on the upper side-edge vortex. For the P3 sensor, the pressure spectrum of the slotted flap side-edge is lower than that of the baseline flap side-edge. For the P4 sensor, the pressure spectrum of the slotted flap side-edge is decreased compared with the baseline flap side-edge.

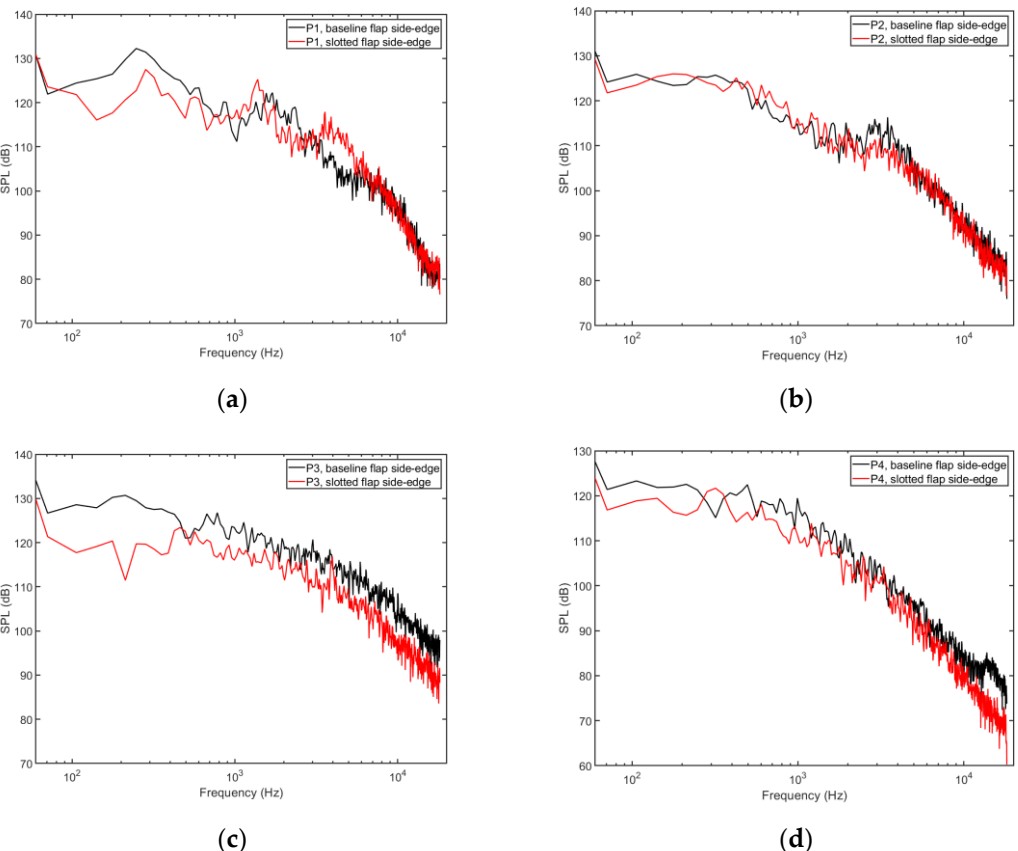

**Figure 12.** The comparison of the pressure fluctuations spectra at four different sensors on the flap side-edge between the baseline flap side-edge and slotted flap side-edge. (**a**) P1; (**b**) P2; (**c**) P3; (**d**) P4.

### 3.2. Farfield Noise Prediction

In this study, the far-field noise of flap side-edge is predicted by the Ffowcs-Williams and Hawkings' equation (FW-H) [35], which can be written in terms of acoustic pressure in the following form:

$$\left(\frac{1}{c^2}\frac{\partial^2}{\partial t^2} - \frac{\partial^2}{\partial x_i^2}\right)p'(\boldsymbol{x},t) = \frac{\partial}{\partial t}[\rho_0 v_n \delta(f)] - \frac{\partial}{\partial x_i}[pn_i(f)] + \frac{\partial^2}{\partial x_i \partial x_j}[H(f)T_{ij}], \quad (1)$$

where $p'$ is the acoustic pressure; $c_0$ and $\rho_0$ are the speed of sound and the free-stream fluid density; $v_n$ is the local normal velocity of a surface; $p$ is the local gage pressure on the surface; $\delta(f)$ and $H(f)$ are the Dirac delta and the Heaviside functions. The three source terms on the right-hand side are known as thickness noise, loading noise, and quadrupole noise.

Formulation 1A of Farassat [36] is a well-known solution of the FW-H equation by neglecting the quadrupole source term as the solid surface moves at a low Mach number speed. The thickness and the loading noise components of Formulation 1A are:

$$p'_T(\boldsymbol{x},t) = \frac{1}{4\pi}\int_{f=0}\left[\frac{\dot{v}_n}{r(1-M_r)^2}\right]_{ret}\rho_0 dS + \frac{1}{4\pi}\int_{f=0}\left[\frac{\dot{v}_n\left(r\dot{M}_r + c_0\left(M_r - M^2\right)\right)}{r^2(1-M_r)^3}\right]_{ret}\rho_0 dS, \quad (2)$$

$$p'_L(\boldsymbol{x},t) = \frac{1}{4\pi}\int_{f=0}\left[\frac{\dot{L}_r}{c_0 r(1-M_r)^2}\right]_{ret}dS + \frac{1}{4\pi}\int_{f=0}\left[\frac{(L_r - L_M)}{r^2(1-M_r)^2}\right]_{ret}dS +$$
$$\frac{1}{4\pi}\int_{f=0}\left[\frac{L_r\left(r\dot{M}_r + c_0\left(M_r - M^2\right)\right)}{c_0 r^2(1-M_r)^3}\right]_{ret}dS, \quad (3)$$

where

$$M_r = M_i \hat{r}_i, \quad \dot{M}_r = \dot{M}_i \hat{r}_i$$
$$L_r = L_i \hat{r}_i, \quad \dot{L}_r = \dot{L}_i \hat{r}_i, \quad L_M = L_i M_i, \quad L_i = p n_i, \tag{4}$$

In the preceding equations, $p'_T$ and $p'_L$ denote the acoustic pressure due to thickness and the loading noise source, respectively. $x$ is the observer position and $t$ is the observer time. The dot over a variable implies source-time differentiation of that variable, $r$ is the distance of the source point to the observer position $x$, $\hat{r}_i$ is the component of the unit radiation vector, $M_i$ is the component of the Mach number of the source in the radiation direction at the emission time, $n_i$ is the component of the outward surface normal vector. The subscript *ret* denotes that the integrand is evaluated at the retarded time.

Figure 13 shows the computed flap side-edge spectra of the baseline flap side-edge. Both approaches, namely the delayed time approach and the advanced time approach [37], have been applied for cross-validation. It can be seen clearly that identical spectra have been obtained by the two different approaches. This is a verification of the FW-H solver. The spectra of the flap side-edge noise are clearly of a broadband nature, consistent with prior experimental observations for flap side-edge noise [4,17]. Moreover, two broadband humps can be observed. The first broadband hump is around the frequency of 250 Hz. The corresponding Strouhal number is $St = fL/U_0 = 1.8$ based on the flap chord and the incoming velocity. The second broadband peak is close to 2000 Hz, which is about 10 times the peak frequency of the first broadband hump. The corresponding Strouhal number for the peak frequency of the second broadband hump is 1.4 based on the flap thickness and the incoming velocity, which is consistent with experiment measurement [4,6,16]. The spectra at low frequencies are proportional to the frequency squared, and the spectra falloff above the peak frequency of the first hump is proportional to the inverse of the second power of the frequency. At the high-frequency range, the spectra falloff is proportional to the inverse of the fourth power of the frequency. These characteristics of the flap side-edge noise spectra are consistent with the prior experimental observations [4,6,16].

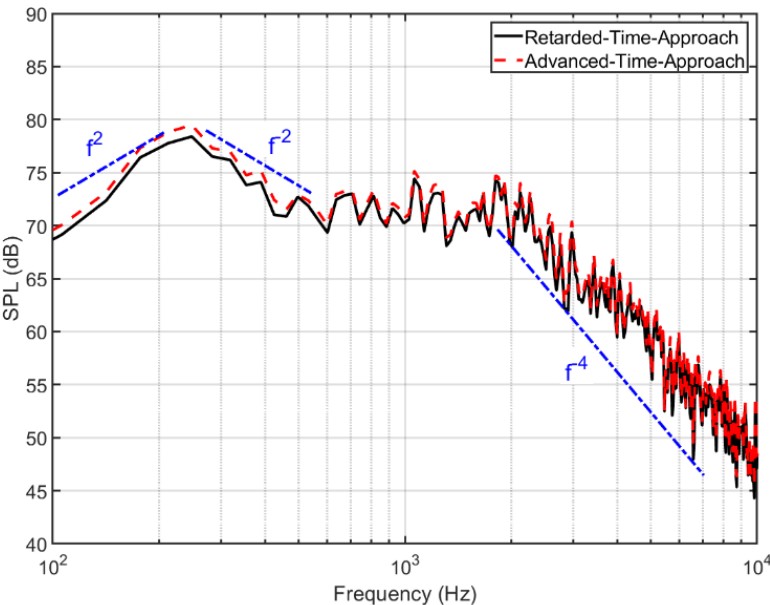

**Figure 13.** The noise spectra of the baseline flap side-edge.

In order to compute the directivity of the flap side-edge noise, 36 microphones are placed around the flap side-edge, as shown in Figure 14. The distance from the center of the flap to the far-field microphones is 8 m. Figure 15 compares the directivity of the overall sound pressure level (OASPL) of the flap side-edge noise between the baseline flap side-edge and the slotted flap side-edge. It can be seen clearly that the directivity of the

baseline flap side-edge is dipole-like. The orientation of the dipole is perpendicular to the chord of the flap side-edge. This is in agreement with the experiment, measurement, and theory [6,16]. In contrast, the slotted flap side-edge noise has complex multipole directivity.

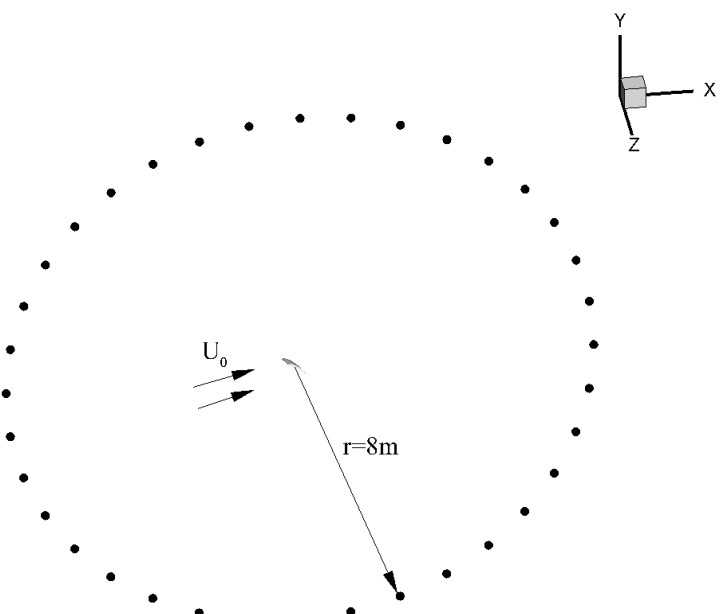

**Figure 14.** The locations of the far-field microphones.

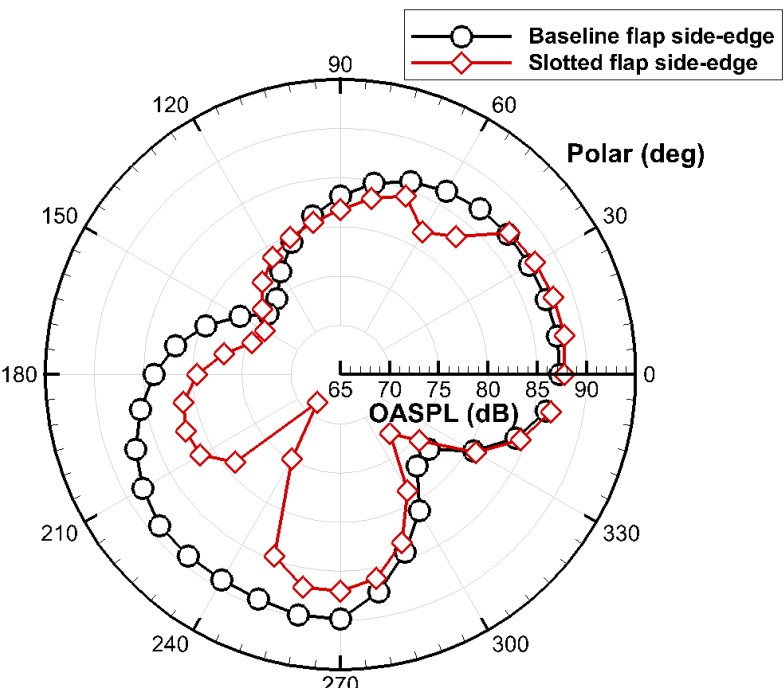

**Figure 15.** The comparison of the directivity of the flap side-edge noise between the baseline flap side-edge and the slotted flap side-edge.

Compared with the baseline flap side-edge, the slotted flap side-edge can reduce the flap noise significantly, especially at polar angles of 180 to 300 degrees. The maximum noise reduction can be achieved at about 20 dB at a polar angle of 230 degrees. In addition, the slotted flap side-edge noise level is a little higher than the baseline flap side-edge at

several polar angles. Considering the most important observation angles are from 180 to 360 degrees, the slotted flap side-edge is an effective noise reduction technology.

Additionally, the flap side-edge noise spectra are compared. Figure 16 compares the far field noise spectra between the baseline flap side-edge and slotted flap side-edge at four different polar angles. Figure 16a shows the comparison of the flap side-edge noise spectra at a polar angle of 180 degrees. It can be seen that the slotted flap side-edge noise spectrum is lower than the spectrum of the baseline flap side-edge over the whole frequency range. The noise reduction is significant, especially in the low- and high-frequency ranges. An identical trend can be found at the polar angle of 210 degrees, as shown in Figure 16b. As the polar angle increased to 270 degrees, the difference in flap side-edge noise between the baseline flap side-edge and the slotted flap side-edge was quite small, as shown in Figure 16c. At the polar angle of 310 degrees, the slotted flap side-edge noise spectra are lower than the spectra of the baseline flap side-edge only in the low-frequency range.

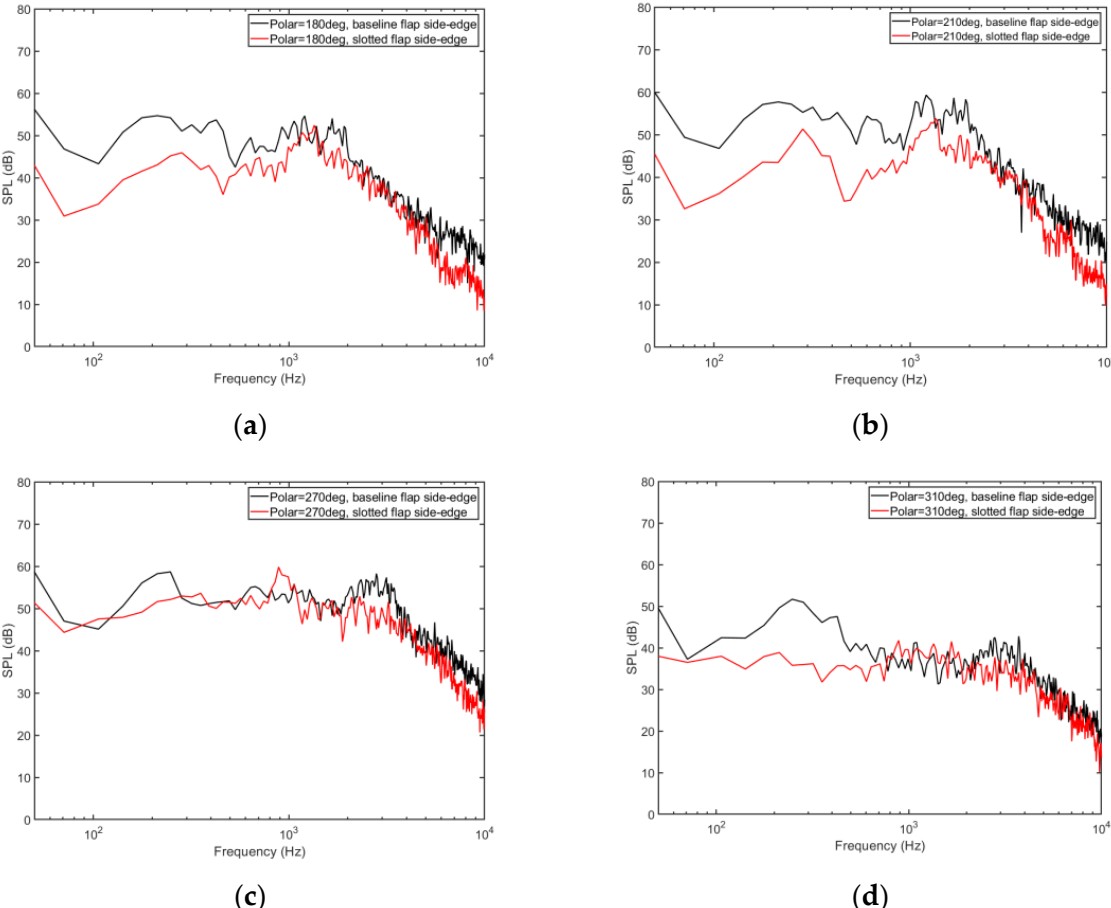

**Figure 16.** Noise comparison between the baseline flap side-edge and slotted flap side-edge. (**a**) Polar angle of 180 degrees; (**b**) polar angle of 210 degrees; (**c**) polar angle of 270 degrees; (**d**) polar angle of 310 degrees.

Figure 17 compares the source power level spectra between the baseline flap side-edge and the slotted flap side-edge. It can be clearly seen that the slotted flap side-edge noise source is relatively lower in the low- and high-frequency ranges compared with the baseline flap side-edge noise. In contrast, the noise in the middle-frequency range of about 900 Hz is increased. In general, the overall PWL is reduced by about 3.3 dB by the slotted flap side-edge.

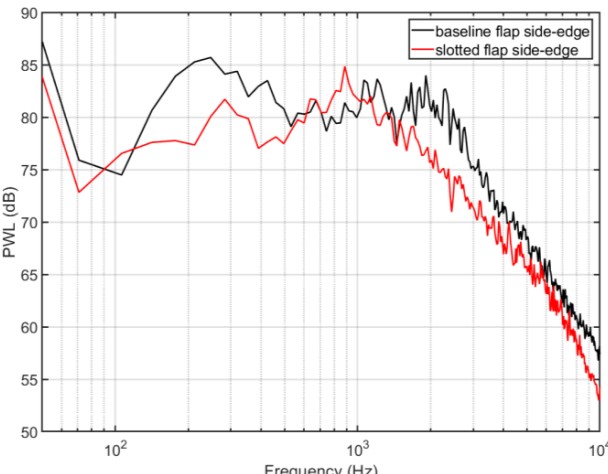

**Figure 17.** The comparison of the sound power level (PWL) between the baseline flap side-edge and the slotted flap side-edge.

## 4. Conclusions

In the past decades, advances in the aero-engine noise suppression technique have significantly reduced aircraft aero-engine noise. Airframe noise now represents a dominant noise source, especially at the landing stage. One of the major airframe noise sources is the flap side-edge in high-lift systems.

In this study, a novel flap side-edge noise reduction method was proposed to reduce the flap side-edge noise based on the concept of active blowing air. However, the active blowing air technique requires an extra air source and control system, which makes the system quite complex. Therefore, we proposed a novel passive blowing air control method that opened a long slot from the lower surface to the tip surface of the flap. A secondary jet flow would be produced in the long slot due to the local pressure difference between the lower surface and the tip surface. As a result, the passively forced secondary jet flow would reduce the flap side-edge noise.

In order to evaluate this newly proposed slotted flap side-edge noise reduction method, the LBM solver PowerFLOW was applied for the unsteady flow field computation, and the far-field noise was predicted by the FW-H equation. It is demonstrated that the dominant flow features of the flow at the flap side-edges are the double-vortex structure. Compared with the baseline flap side-edge, the newly proposed slotted flap side-edge can reduce the flap side-edge noise significantly at the observation point below the flap.

The flow fields around the baseline flap side-edge and the new slotted flap side-edge were compared and analyzed to reveal the underlying noise reduction mechanisms. It is demonstrated that the new noise reduction method can achieve a noise reduction of about 3.3 dB over the whole frequency range. The noise reduction mechanism has been revealed. On the one hand, it is found that the secondary jet flow can dissipate the original flap side-edge vortical structures, which reduces the strength of the vorticity. On the other hand, the secondary jet flow can displace the vortex away from the flap surface, thus weakening the interaction between the flap side-edge vortexes and the flap side-edge surface. As a result, the pressure fluctuations on the flap surface decreased significantly, and the flap side-edge noise was reduced.

**Author Contributions:** Conceptualization, B.B. and P.L.; methodology, Y.Z. and B.B.; software, Y.Z. and B.B.; validation, Y.Z. and B.B.; formal analysis, B.B. and Y.Z.; investigation, Y.Z., D.L. and B.B.; writing—original draft preparation, Y.Z.; writing—review and editing, D.L., B.B. and P.L.; visualization, Y.Z. and B.B.; supervision, D.L. and P.L.; project administration, D.L. and P.L.; funding acquisition, B.B. All authors have read and agreed to the published version of the manuscript.

**Funding:** This research was funded by the National Science Foundation of China, grant number 12102483.

**Institutional Review Board Statement:** Not applicable.

**Informed Consent Statement:** Not applicable.

**Data Availability Statement:** Not applicable.

**Acknowledgments:** Not applicable.

**Conflicts of Interest:** The authors declare no conflict of interest. The funders had no role in the design of the study; in the collection, analyses, or interpretation of data; in the writing of the manuscript; or in the decision to publish the results.

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
