# Peer review of "A Numerical Study on the Flap Side-Edge Noise Reduction Using Passive Blowing Air Concept"

_aerospace, doi:10.3390/aerospace10040360_

Round 1

Reviewer 1 Report

The paper reports a numerical investigation regarding a novel passive control strategy to reduce the noise from flap. The presented solution is a slotted flap. Such tecnique reduce noise emitted by dissipating the flap side-edge vortex and displacing the vortical structures far away from the flap surface. The editor does not recognize particular issue in the manuscript.

Author Response

We have improved our manuscript according to the comments of reviewers. We would like to sincerely thank the reviewer for the very constructive comments and suggestions. We have taken the reviewer’s concerns into consideration and have answered all the questions with corresponding changes to the manuscript.

Reviewer 2 Report

The manuscript reports a numerical work about a passive air blowing concept for noise reduction of flap side-edge noise. The work adopts Powerflow (Lattice Boltzmann method) in combination with Formulation 1A of Farassat, a type of Ffowcs-Williams and Hawking method, to assess the effectiveness of the concept.

While the work shows interesting concepts and results, I cannot recommend its publication due to some issues as follows.

1. Lack of convergence study

Convergence of solutions over mesh is critical to the establishment of the result correctness. However, the manuscript does not contain this discussion, leaving readers not sure about the convergence of the numerical results. 

2. Lack of discussion about its effect on the aerodynamic performance of the airfoil

The manuscript only shows the decreased strength of the vortical structure generated by the flap side-edge. However, it does not mention the effect of the concept on the aerodynamic performance of the airfoil. Quantification of the aerodynamic coefficients can help readers to understand the effectiveness of the concept.

3. Justification of using Formulation 1A of Farassat to determine the acoustics

This formulation is suitable for low Mach number flows as it assumes solid surfaces move in the Mach number range. For aviation, the flow is faster than this range. Thus, it may generate some discrepancies in assessing the acoustics of the airfoil. Could the authors please justify its adoption in this manuscript?

Author Response

Attachment is our response to the reviewer's comments.

Reviewer 3 Report

This paper is concerned with computing the flap side-edge noise under active blowing. A long slot is opened from the lower surface to the tip surface of the flap to induce a secondary jet flow. The unsteady flow field around the flap side-edge was computed by the lattice Boltzmann solver PowerFLOW and the far field noise was predicted by the FW-H equation.  The authors have demonstrated that the dominant features of the flow at the flap side-edges are the double-vortex structures and that the new passive blowing air reduction method can achieve about 5dB noise reduction.

The paper presents a novel idea and I think it is worthy of publications. Some suggestions for improvements:

1, Indicate on the spectra the limit of the validity of your resolutions.

2. Numerical Validation is needed or Comparison with experiment.

3. Show the total power level

4. Explain why the effect is frequency dependent.

5. The explanation to the suppression mechanism can be improved.

Author Response

Attached pdf file is the response to the reviewer's comments.

Round 2

Reviewer 2 Report

I would like to thank the authors taking my concerns into account. The manuscript is in a good shape, so I recommend its publication.

Reviewer 3 Report

The authors have addressed my concern